# Using Iron Ore Ultra-Fines for Hydrogen-Based Fluidized Bed Direct Reduction—A Mathematical Evaluation

**DOI:** 10.3390/ma15113943

**Published:** 2022-06-01

**Authors:** Thomas Wolfinger, Daniel Spreitzer, Johannes Schenk

**Affiliations:** 1K1-MET GmbH, Stahlstraße 14, 4020 Linz, Austria; johannes.schenk@unileoben.ac.at; 2Primetals Technologies Austria GmbH, Turmstraße 44, 4020 Linz, Austria; daniel.spreitzer@primetals.com; 3Ferrous Metallurgy, Montanuniversitaet Leoben, Franz-Josef-Straße 18, 8700 Leoben, Austria

**Keywords:** direct reduction, hydrogen, iron ore ultra-fines, pellet feed, fluidized bed, sticking

## Abstract

This mathematical evaluation focuses on iron ore ultra-fines for their use in a novel hydrogen-based fluidized bed direct reduction process. The benefits of such a process include reduced CO_2_ emissions and energy consumption per ton of product, lower operational and capital expenditure, and a higher oxide yield. Typical samples of iron ore ultra-fines, such as pellet feed, are given and classified for a fluidized bed. An operating field for a hydrogen-based fluidized bed direct reduction process using iron ore ultra-fines is shown in the fluidized state diagram following Reh’s approach and compared to other processes. The effects of the process conditions and the agglomeration phenomenon sticking were analyzed and evaluated with mathematical case studies. The agglomeration phenomenon sticking was identified as the most critical issue; thus, the dependencies of the fluid dynamics on the characteristic diameter were examined.

## 1. Introduction

A new disruptive ironmaking technology, HYFOR^®^, is under development as a measure to achieve the ambitious goals for a carbon-neutral economy [1,2,3,4,5]. The objective is to use iron ore ultra-fines (<150 µm), such as pellet feed, directly in a hydrogen-based fluidized bed direct reduction process. Using hydrogen as reducing gas in a direct reduction process is a promising route to reduce specific CO_2_ emissions [2,6,7,8,9]. Theoretically, emitting 0.1 to 0.25 tons of CO_2_ per ton of crude steel is achievable using the hydrogen generated from electrolysis with renewable energies [9]. Using natural gas as a basis for the reduction process, CO_2_ emissions can be halved compared with the dominant blast furnace–basic oxygen furnace (BF–BOF) route, which emits approximately 1.7 to 1.9 tons of CO_2_ per ton of crude steel [6,9,10,11]. In 2020, about 70% of the world’s steel production of 1.88 billion tons was produced via the BF–BOF route [11,12]. In the same year, only 106 million tons of direct reduced iron (DRI) and hot briquetted iron (HBI) were produced [12].

Besides the need to reduce the CO_2_ emissions in the ironmaking process, a further decrease is possible because of the omission of the agglomeration process (e.g., pelletizing) when using iron ore ultra-fines directly [2,6,13]. The benefits of HYFOR^®^ technology are summarized as follows: low CO_2_ footprint, low operational and capital expenditure as well as overall energy consumption per ton of product, and high iron oxide yield due to the possibility of recycling oxide dust and using the undersized fraction as a feed material [4,5].

With similar advantages, the Finmet^®^ and the Circored^®^ technologies have been identified as direct reduction technologies that use hydrogen-rich reducing gas and iron ore fines directly. In these processes, the feed material is coarser, with a particle size distribution for the Finmet^®^ process between 0.05 and 8 mm, sinter feed ore, and for the Circored^®^ process, a size distribution between 0.1 and 2.0 mm [11,14,15,16,17,18]. For the Circored^®^ process, the particle size distribution must be adjusted in a pre-step before the actual process. Oversized sinter feed ore needs to be crushed, and undersized pellet feed ore needs to be micro-granulated [11,17,18]. Hence, the particle size distribution of the feed material is of significant importance.

To achieve the advantages of HYFOR^®^ technology, challenges need to be solved in the development of the process, mainly to keep the fluidization stable throughout the transformation to metallic iron. The main fluidization problem reported is the agglomeration phenomenon sticking, which hinders stable fluidization [19,20,21,22,23,24,25,26,27,28,29,30,31,32,33,34,35,36,37,38], although most studies restrict the feed material of ultra-fine ore to defined size fractions, e.g., Zang et al., 106–150 µm [35]; He et al., approximately 20–200 µm [23]; Du et al., 74–150 µm [37]; Lei et al., 106–150 µm [36]; and Zhong et al., different size fractions between 50 and 200 µm [26,32,33,34].

The following sections examine the usability and challenges for using iron ore ultra-fines in a fluidized bed direct reduction process in a mathematical context. First, typical samples of iron ore ultra-fines are introduced and classified for a fluidized bed. Second, the fluid dynamics of the hydrogen-based fluidized bed direct reduction process using iron ore ultra-fines are analyzed using the fluidized state diagram following Reh’s approach. Third, to evaluate the influence of the process conditions and the agglomeration phenomenon sticking during fluidized bed reduction, two mathematical case studies are outlined, one with and one without sticking.

## 2. Materials and Methods

The materials focused on were iron ore ultra-fines, such as pellet feed, and they were first characterized regarding the material properties relating to fluidization. Second, the necessary background for classifying such powders and the formulas for the mathematical evaluation were given.

### 2.1. Iron Ore Ultra-Fines

The main characteristic of iron ore ultra-fines is their particle size distribution of <150 µm, as shown in Figure 1. for some typical iron ore ultra-fines. The distribution course was similar for all five samples due to the beneficiation of the iron ores.

Table 1 summarizes the material properties relating to the fluidization of the iron ore ultra-fines given and the chemical analysis. The bulk densities were measured using an Erweka SVM 200, ERWEKA GmbH, Langen, Germany (250 mL glass cylinder). The particle densities were, in this case, close to the true densities because of the beneficiated ultra-fine materials, which were measured with a helium pycnometer, Quantachrome Instruments, Boynton Beach, FL, USA (Quantachrome Ultrapycnometer 1000). The particle density was lower than the true density because open cracks and pores on the particle’s surface were included in the particle volume. Despite this, the measurement was easy to apply, and there was little deviation due to the intensive beneficiation of the ultra-fine material. The void fraction of a packed bed is calculated from the bulk density, $\rho_{\mathrm{Bulk}}$, and particle density, $\rho_{p}$, and is defined as follows:(1)$$\varepsilon =1\;-\frac{\rho_{\mathrm{Bulk}}}{\rho_{p}}$$
where the true density is used for the particle density. The void fraction of the bed material in the fluidized state, $\varepsilon_{f}$, was similarly calculated, but by using the density of the fluidized bed instead of the bulk density. The particle size distributions of the iron ore ultra-fines, Figure 1, were measured using laser diffraction measurement, CILAS, Orléans, France (CILAS type 1064 L). The characteristic diameters d10, d50, and d90 and the maximum particle size in Table 1 were obtained from these deduced particle size distributions. The sphericity, ϕ, was estimated according to the schemata by Krumbein and Sloss based on scanning electron microscopy pictures taken with a Quanta 200 Mk2, FEI Company, Hillsboro, OR, USA [39]. An example of iron ore ultra-fines is given in Figure 2 from sample E, which shows that the particles were irregular and flaky with smooth surfaces. According to the chemical analysis, the iron ore ultra-fines showed, after beneficiation, a high iron and a low gangue content. The high FeO content indicated that the iron ore ultra-fines were magnetite iron ores.

### 2.2. Correlations for the Evaluation of Using Iron Ore Ultra-Fines in a Fluidized Bed

#### 2.2.1. General Classification Diagram of Fluidized Particles

Geldart classified the fluidization behavior of solid powder materials into four groups within a plot of density difference between the particles and the fluidizing medium $\left(\rho_{p}-{\;\rho }_{f}\right)$ against the mean particle size $d_{p}$. Group A particles are fine and aeratable solids, which show high bed expansion and good fluidization quality. Group B particles are intermediate-sized particles that show little bed expansion, and bubbles appear when fluidization begins. Group C particles are very fine and cohesive materials that form channels and agglomerates and are difficult to fluidize. Group D particles are coarse and large solids that form vertical channels. The boundaries between these groups were experimentally determined under ambient conditions using air as a fluidization gas [40]. Goossens used the dimensionless Archimedes number, Ar, to divide between the groups, which is given as follows:(2)$$\mathrm{Ar}=\frac{d_{p}^{3}{\;\times \;\rho }_{f}\;\times \;\left(\rho_{p}-{\;\rho }_{f}\right)\times \;g}{\eta_{f}^{2}}$$
where $d_{p}$ is the mean particle size, $\rho_{f}$ is the density of the fluidizing medium, $\rho_{p}$ is the particle density, g is the gravity (9.81 m/s^2^), and $\eta_{f}$ is the dynamic gas viscosity. The boundaries between groups C, A/C, A, B, and D were defined by Archimedes numbers of 0.97, 9.8, 88.5, and 176,900, respectively, using laminar-to-turbulent ratios for the conditions of entrainment and incipient fluidization [41].

#### 2.2.2. Fluidized State Diagram Following Reh’s Approach

This diagram is used to classify different reactor types according to the material and gas properties and process conditions, for example, shaft reactors or circulating fluidized beds. For the calculation, Equations (3)–(7) were used, including the dimensionless numbers the Archimedes number Ar, Reynolds number Re, modified Froude number ${\mathrm{Fr}}^{*}$, Liatschenko number M, and the drag coefficient. The drag coefficient for a single particle, $C_{D}$, was calculated according to the empirically found formula established by Haider and Levenspiel [42]. The adapted drag coefficient to account for mutual interactions between particles for homogenous fluidization, CDRe, ε, was given by the empirically found formula proposed by Richardson and Zaki [43]. For heterogenous fluidization, the drag coefficient has to be interpolated towards the value of one.
(3)Re=dp × ρf × uslipηf
(4)CD=24Re × 1 + 8.1716 × e−4.0655 × ϕ × Re0.0964+0.5565 × ϕ+73.9 × e−5.0748 × ϕ × ReRe+5.378 × e6.2122 × ϕ
(5)CDRe, ε=CDεα
(6)Fr*=34 × uslip2g × dp × ρfρp− ρf=34 × Re2Ar=nCDRe, ε
(7)$$M=\frac{u_{\mathrm{slip}}^{3}{\;\times \;\rho }_{f}^{2}}{{g\;\times \;\eta }_{f}\;\times \;\left(\rho_{p}-{\;\rho }_{f}\right)}$$

The slip velocity, $u_{\mathrm{slip}}$, is given by the superficial gas velocity in the reactor, u, divided by the void fraction of the bed material in the fluidized state, $\varepsilon_{f}$. The load factor, n, defines the ratio of upwards to downwards acceleration, and index α is used to account for the mutual interactions between particles. The modified Froude number is defined by the ratio between the inertia force and the weight force, which can be described using the Reynolds and Archimedes numbers, given in Equation (6). The link between the modified Froude number and the adapted drag coefficient is given via the Reynolds and Archimedes numbers resulting from the balanced gravitational force, buoyancy force, and drag force on the particles.

#### 2.2.3. Minimum Fluidization Velocity $u_{\mathrm{mf}}$

At the point of incipient fluidization, $u_{\mathrm{mf}}$ can be calculated by inserting and rearranging Equation (8) for pressure loss through the packed bed from Ergun in Equation (9), when the pressure loss across the bed is balanced by the effective weight of the bed:(8)$$\frac{∆P}{H}{\;=\;K}_{l}\;\times \;\frac{{\left(1\;-\;\varepsilon \right)}^{2}{\;\times \;\eta }_{f}{\;\times \;u}_{\mathrm{mf}}}{\varepsilon^{3}\;\times \;\phi^{2}{\;\times \;d}_{p}{}^{2}}{\;+\;K}_{t}\;\times \;\frac{\left(1\;-\;\varepsilon \right){\;\times \;\rho }_{f}{\;\times \;u}_{\mathrm{mf}}^{2}}{\varepsilon^{3}\;\times \;\phi {\;\times \;d}_{p}}$$
(9)$$\frac{∆P}{H}=\left(1\;-\;\varepsilon \right)\;\times \;\left(\rho_{p}-{\;\rho }_{f}\right)\;\times \;g$$
(10)$$u_{\mathrm{mf}}=\frac{-\frac{K_{l}\;\times \;\left(1\;-{\;\varepsilon }_{\mathrm{mf}}\right)}{\varepsilon_{\mathrm{mf}}^{3}\;\times \;\phi^{2}}\;\times \;\frac{d_{p}{\;\times \;\rho }_{f}}{\eta_{f}}+\sqrt{{\left(\frac{K_{l}\;\times \;\left(1\;-{\;\varepsilon }_{\mathrm{mf}}\right)}{\varepsilon_{\mathrm{mf}}^{3}\;\times \;\phi^{2}}\;\times \;\frac{d_{p}{\;\times \;\rho }_{f}}{\eta_{f}}\right)}^{2}-\;4\;\times \;\frac{K_{t}}{\varepsilon_{\mathrm{mf}}^{3}\;\times \;\phi }\;\times {\left(\frac{d_{p}{\;\times \;\rho }_{f}}{\eta_{f}}\right)}^{2}\;\times \;\left(-\frac{d_{p}^{3}{\;\times \;\rho }_{f}\;\times \;\left(\rho_{p}-{\;\rho }_{f}\right)\;\times \;g}{\eta_{f}{}^{2}}\right)}}{2\;\times \;\frac{K_{t}}{\varepsilon_{\mathrm{mf}}^{3}\;\times \;\phi }\;\times {\left(\frac{d_{p}{\;\times \;\rho }_{f}}{\eta_{f}}\right)}^{2}}$$
where ∆P is the pressure drop caused by gas flow through a packed bed, H is the height of the packed bed, $\varepsilon_{\mathrm{mf}}$ is the void fraction of the bed material at the point of incipient fluidization, and K_l_ and K_t_ are constants. Both constants K_l_ and have to be determined experimentally, especially for non-spherical particles.

## 3. Results and Discussion

### 3.1. Classification of Iron Ore Ultra-Fine Powders for a Fluidized Bed

Based on the classification according to Geldart and Goossens, the five samples of iron ore ultra-fines from Table 1 are illustrated in Figure 3. The dots represent the particle diameter, $d_{i}$, for every 10 vol.−% underflow interval of the particle size distribution, meaning the first dot d10 at 10 vol.−%, second dot d20 at 20 vol.−% underflow, etc., and the last dot d100 at 100 vol.−%. This form of illustration was chosen since the particle size distribution is not wholly represented by the mean particle diameter alone.

The broad particle size distribution of iron ore ultra-fines ranged from Group C over Group A/C to Group A materials and made it difficult to forecast their fluidization behavior, although one assumption of the boundaries, according to Goossens, did not fit for iron ore ultra-fines. For boundary A/B, a void fraction of 0.383 at minimum fluidization velocity was taken, which was too low for iron ore ultra-fines, given the loose bulk density between 1800 and 2200 kg/m^3^. A true density in the range of 4800 to 5100 kg/m^3^ resulted in a void fraction of 0.5 to 0.65, meaning that the A/B boundary was reduced from Ar = 88.5 to 25 and below, and thus moved to smaller particles. Similar classification maps using slightly different assumptions can be found in the literature [44,45,46], though by omitting the viscosity of the fluidizing medium, such a classification has only a limited scope of application. Thus, a transformation to other fluids or process conditions is only possible to a limited extent.

Focusing on the hydrogen-based reduction process, only the one proposed by Goossens had validity. Goossens proposed that the boundaries between groups C, A/C, A, B, and D are Archimedes numbers of 0.97, 9.8, 88.5, and 176,900, respectively. Under the given initial conditions, the particle diameter to be within group A, which showed good fluidization quality, was between 120 and 250 or 165 µm, considering an Archimedes number of 88.5 or 25, respectively, for the boundary between groups A and B, meaning that the fluidization behavior of the materials remained similar to that of group C and A/C, and hardly changed during the course of reduction. This statement is only valid when the superficial gas velocity remains constant. An estimation of the influence of the particle size distribution is, with this classification, not feasible.

### 3.2. Operating Field of Iron Ore Ultra-Fines for a Fluidized Bed Reactor

One characteristic of iron ore ultra-fines is the broad particle size distribution. To account for approximately 95 wt.−% of the total mass, the size range is primarily between 2 and 90 µm. For the process data, the temperature range was between 873 and 1173 K [4]. The pressure was defined with a 0.1 bar gauge to be slightly above ambient. The gas velocity followed Spreitzer and Schenk between 0.15 and 0.30 m/s for hydrogen mixtures with the extreme case of up to 80% H_2_O [29,47]. Figure 4 shows an extended version of the Reh diagram, including various gas–solid reactor systems (fields a to e), the FINMET^®^ operating field, and the operating field R for the reduction of iron ore ultra-fines in a fluidized bed. The operating field R is circumscribed as follows:1.Particle diameters, $d_{P}$, in size range of 2 to 90 µm, to account for 95 wt.−%;2.Densities of the ore and DRI, $\rho_{p}$, between 5000 and 3500 kg/m^3^;3.Gas densities, $\rho_{f}$, and dynamic viscosities, $\eta_{f}$, for the H_2_ and H_2_/H_2_O mixtures at temperatures of 873 to 1123 K and a pressure of 0.1 barg;4.Superficial gas velocity, u, between 0.15 and 0.30 m/s.

In contrast to the operating field for iron ore ultra-fines is the operating field for sinter feed ore, mainly across the bubbling and circulating field [15,16]. Several issues need to be discussed, focusing only on iron ore ultra-fines as a feed material. First, commonly drawn fluidized state diagrams following Reh’s approach are only illustrated to the lower limit of Re = 1.0 × 10^−1^ [16,48,49]. An extended version of the fluidized state diagram following Reh’s approach is necessary. Second, only fluid dynamic forces are considered. According to many authors, such a fine material shows interparticle forces, influencing the fluidization behavior [46,50,51,52,53,54,55,56,57,58,59]. Third, it is impossible to distinguish between the different fluidization behaviors introduced by Geldart [40], especially for Group C particles, which are difficult to fluidize and show channels and agglomerates, rather than fluidization behavior. According to Figure 3, at density differences of 3500 to 5000 kg/m^3^, a particle diameter of 20 µm is the boundary for Group C particles, resulting in an Archimedes number of around 1.0 × 10^−2^, and, therefore, for half of the operating field. Fourth, secondary parameters, such as the geometry of the reactor or the gas distributor, the bed-height-to-diameter ratio, or an uneven particle size distribution, are not considered.

When processing iron ore ultra-fines in a fluidized bed reactor, a fluidization regime diagram regarding the operating field of such a process must account for the following variables: classification of the fluidization behavior according to Geldart’s groups; interparticle forces; heterogenous/aggregative fluidization; bed-height-to-diameter ratio; a non-simplified minimum fluidization velocity; entrainment velocity; feed material properties, such as the particle size distribution; and fluidizing medium properties, such as the density and viscosity.

### 3.3. Mathematical Case Studies

The extended version of the fluidized state diagram following Reh’s approach was used to illustrate and interpret the two mathematical case studies. The first case study was calculated for a fluidized bed direct reduction process using iron ore ultra-fines if no agglomeration of the particles occurred, meaning without sticking. The second case study considered the agglomeration phenomenon sticking by means of increasing the particle diameters.

#### 3.3.1. Assumptions of the Reduction Process for the Mathematical Case Studies

The influence of the reduction process was evaluated for materials with reduction degrees (RDs) of 0%, 30%, 70%, and 95%, meaning that, at 0%, the pre-heated material was charged into the reactor. The value of 95% indicated the end of the reduction process, and 30% and 70% were in between. The degree of reduction was defined via oxygen removal from the iron oxide, as follows:(11)$$\mathrm{RD}=\left(1\;-\frac{O}{1.5\;\times \;\mathrm{Fe}}\right)\;\times \;100\%$$
where O and Fe are the amounts of oxygen and iron of the sample in mol. A RD of 0% represents hematite, Fe_2_O_3_; 11% magnetite, Fe_3_O_4_; 33% wüstite, FeO; and 100% metallic iron only, Fe. Upper and lower temperature limits of 1173 and 873 K were chosen according the publication of the HYFOR^®^ technology [4]. The mass and gas flow were unknown at this point, and the gas utilization to account for kinetic limitations was assumed following data from the literature [20,29,31,47]. The gas utilization, ζ, describes the deviation from the theoretically possible conversion of the gas from the reduction reaction. High gas utilization of 80% for RDs of 0%, 30%, and 70%, and low gas utilization of 20% for RD 95% were assumed. Hence, the gas composition was given by multiplying the thermodynamically resulting gas composition at the given temperature by the set gas utilization. The pressure was assumed with a 0.1 bar gauge to account for a pressures slightly higher than ambient. For a bed density in the fluidized state of 1100 kg/m^3^, the resulting void fraction was 0.78; thus, a superficial gas velocity of 0.25 m/s was assumed, leading to a slip velocity of 0.321 m/s. The superficial gas velocity of 0.25 m/s was chosen, following data from Spreitzer and Schenk [29,47]. It has to be considered that both the particle density and the bulk material density decreased, and the void fraction and the slip velocity thus remained the same. The resulting gas and particle properties are listed in Table 2, though changing bulk material properties, such as the particle size distribution and bulk density, were not considered.

Accounting for the thermodynamic aspect, the whole process’ reduction temperature and gas composition can be calculated using the Baur–Glaessner diagram. In this diagram, the stability areas of the iron oxides are given by the temperature against the gas oxidation degree (GOD). For reduction with hydrogen, the GOD is defined by the gas composition as the ratio of water vapor to hydrogen and water vapor. FactSageTM 7.3 (Database: FactPS, FToxide) was used as the thermodynamic data basis to calculate the stability areas. Each step is drawn in the Baur–Glaessner diagram for both temperatures and the resulting gas compositions in Figure 5. The actual course of the reduction procedure using temperature and the composition of the off-gas has to be in between.

#### 3.3.2. First Case Study: Fluidized Bed Reduction without Sticking

In Table A1 and Table A2, the dimensionless numbers for each reduction degree at 873 and 1173 K, respectively, are presented, and then calculated and drawn into an extended version of the fluidized state diagram according to Reh in Figure 6 [48]. The characteristic particle diameter and sphericity were assumed to remain the same over the whole reduction progress. The gas densities and dynamic gas viscosities were calculated using the process conditions from Table 2. The particle densities at the defined degrees of reduction were calculated assuming that the initial true density was 5000 kg/m^3^, consisting of hematite or oxidized magnetite with 5 wt.−% gangue material. In addition, according to Ergun, the minimum fluidization velocities were calculated using a void fraction at the point of minimum fluidization of 0.65 [60].

As illustrated in Figure 6, the changing process conditions, mainly gas properties, hardly affected fluidization. The dimensionless numbers insignificantly varied with increasing RD. A similar picture was given by the minimum fluidization velocities, which remained, for all conditions, between 0.0017 and 0.0026 m/s, and thus significantly below the superficial gas velocity.

#### 3.3.3. Second Case Study: Fluidized Bed Reduction with Sticking

The same process conditions were chosen for the calculation of the second case study. However, in contrast to the first case study, the characteristic particle diameter and the sphericity increased from the initial 25 to 500 µm and 0.70 to 0.85, respectively, to account for agglomeration. In addition, the density of the agglomerate decreased further because the agglomerate contained spaces between the adhered particles. Hence, a further decrease in the agglomerate density of 30% was assumed. The diameter increase was assumed to start above 33% RD, when the first metallic iron was present. Consequently, at 0% and 30% RD, a diameter of 25 µm, and at 70% and 95% RD, diameters of 250 and 500 µm, respectively, were given. The corresponding data and results are given in Table A3 and Table A4 and are illustrated in Figure 7, showing the effect of increasing diameter.

In contrast to the first case study, the packed bed situation prevailed due to the agglomeration at 70% and 95% RD, considering a void fraction of 0.6 for the packed bed, as listed in Table 1. The gas properties were similar to those in the first case study. The calculation of the minimum fluidization velocity confirmed the packed bed situation at 95% RD and 873 K with 1.06 m/s, taking a superficial gas velocity of 0.25 m/s into account. At 70% RD, the minimum fluidization velocity with 0.24 m/s was close to the superficial gas velocity, which, for a stable fluidization process, is not recommended. The minimum fluidization velocity at 1173 K at 70% RD with 0.17 m/s was below the superficial gas velocity, but at 95% RD with 0.82 m/s, the packed bed situation prevailed.

Hence, the primary influence of fluidization is the particle or agglomerate diameter. Changing gas properties due to the reaction progress only exerted a minor effect, as long as the iron morphology was not altered [37,61,62]. Zhang et al. reported the reduction in fine iron ore in reducing gas mixtures of CO and H_2_ defluidization after a given time when the agglomeration phenomenon sticking occurred [35]. He et al. established a fluidization regime diagram for the reduction of fine iron ore concentrates (mean particle size of 87 µm) with hydrogen in a conical fluidized bed to identify the superficial gas velocities needed to avoid defluidization at a given reduction temperature. They observed that, for agglomerates with a size of approximately 200 to 400 µm, gas velocities above 1.2 m/s are necessary to avoid defluidization [23]. Similar studies with iron powders under a reducing or inert atmosphere and elevated temperatures showed agglomerates of several 100 µm and accompanying fluidization difficulties, such as defluidization [26,32,33,34,36]. These findings confirm the calculations in this work.

The effects of the parameter on the minimum fluidization velocity were given by sensitivity analysis using Equation (10). The initial values for the calculations were taken from the mathematical case at 873 K and 0% RD. The data for the analysis are listed in Table A5 and illustrated in Figure 8 as a deviation from the initial data. It can be seen that the sphericity, the characteristic diameter, and especially the void fraction at $u_{\mathrm{mf}}$ had a significant influence on the minimum fluidization velocity, according to Ergun. In addition, the sphericity and the characteristic diameter affected the minimum fluidization velocity the same way, as indicated by the overlapping curves in Figure 8. A change of 25% in the sphericity and the void fraction at $u_{\mathrm{mf}}$ was the upper limit of the possible deviation. In contrast, an increase in the particle diameter of 25% was, in view of agglomeration phenomena, at the lower limit; increasing the particle diameter from 25 to 250 and 500 µm resulted in increases in the particle diameter of 900% and 1900% and, hence, deviations in $u_{\mathrm{mf}}$ of approximately 10,000% and 40,000%, respectively. This relation explains the strong increase in the minimum fluidization velocity from the first to the second case study for 70% and 95% RD.

Focusing on the primary influencer of the fluid dynamics, the particle diameter, Figure 9 shows the deviations in Ar, Re, Fr*, and $u_{\mathrm{mf}}$ as a function of the change in the particle diameter. The scale of the Y axis for Ar and $u_{\mathrm{mf}}$ was adapted for the factors 500 and 20, respectively. The initial parameter used for the calculation remained similar. The slip velocity for Re and Fr* calculation was set at 0.321 m/s, assuming a superficial gas velocity of 0.25 m/s and a void fraction of 0.78 in the fluidized state, similar to the mathematical case studies. As given in Equations (2) and (3), the Reynolds number was directly proportional to the particle diameter and the Archimedes number by the power of three. This correlation meant an increase from 25 to 250 and 500 µm, and resulted in Reynolds numbers that were 10 and 20 times higher, but Archimedes numbers that were 1000 and 8000 times higher. As a result, the modified Froude number decreased, and the drag coefficient increased similarly to the Reynolds number, as given in Equation (7). Identical to the Archimedes number, the minimum fluidization velocity, according to Ergun, increased by more than a directly proportional amount. This correlation appeared differently in the sensitivity analysis of $u_{\mathrm{mf}}$ in Figure 8, because only a minor increase in the particle diameter is shown, meaning that a larger increase in the particle diameter is represented and, thus, a larger deviation of $u_{\mathrm{mf}}$.

## 4. Conclusions

The novel idea to use iron ore ultra-fines directly in a hydrogen-based fluidized bed direct reduction process enables benefits such as reducing CO_2_ emissions and overall energy consumption, although technical challenges need to be solved in the process development to achieve these advantages. The direct use of iron ore ultra-fines in a fluidized bed was analyzed using the powder classification for a fluidized bed. The reduction progress of iron ore ultra-fines for the hydrogen-based fluidized bed was analyzed using the fluidized state diagram according to Reh. The influence of changing process parameters during hydrogen-based reduction and the meaning of the agglomeration phenomenon sticking on the fluidizing process were evaluated via mathematical case studies. The following conclusions can be drawn:According to Geldart’s and Goossens’s classification, iron ore ultra-fines are mainly Group A, Group A/C, and Group C materials under ambient conditions and air. For the hydrogen-based reduction at higher temperatures, iron ore ultra-fines are mostly Group A/C and Group C, according to Goossens’s classification;The operating field for iron ore ultra-fines needs an extended version of the Reh diagram, and it is not positioned within the general fields of circulating or bubbling fluidized beds;Changing process conditions, such as the temperature and gas properties, hardly affects the fluidization conditions or the minimum fluidization velocity;Changing the characteristic diameter due to sticking significantly affects the fluidization conditions and the minimum fluidization velocity. Thus, the characteristic particle or agglomerate diameter is the most critical parameter for a stable fluidized bed direct reduction process.

## Figures and Tables

**Figure 1 materials-15-03943-f001:**
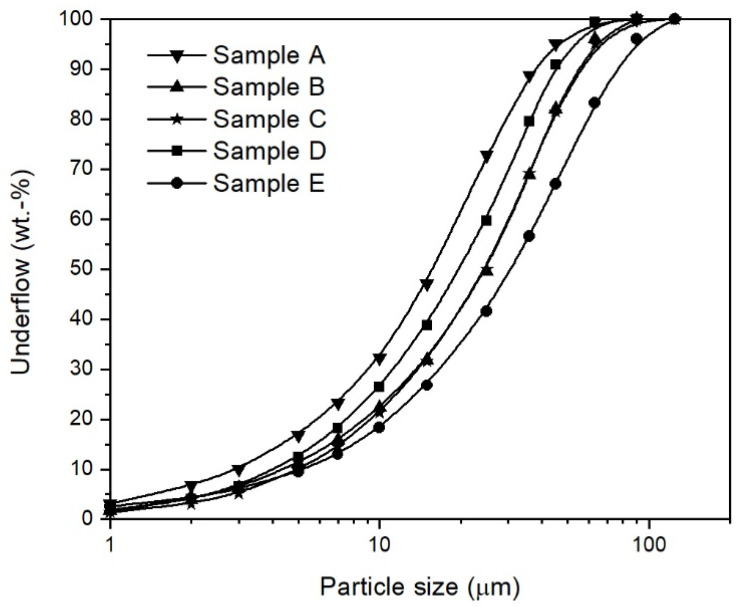
Particle size distributions of typical iron ore ultra-fines.

**Figure 2 materials-15-03943-f002:**
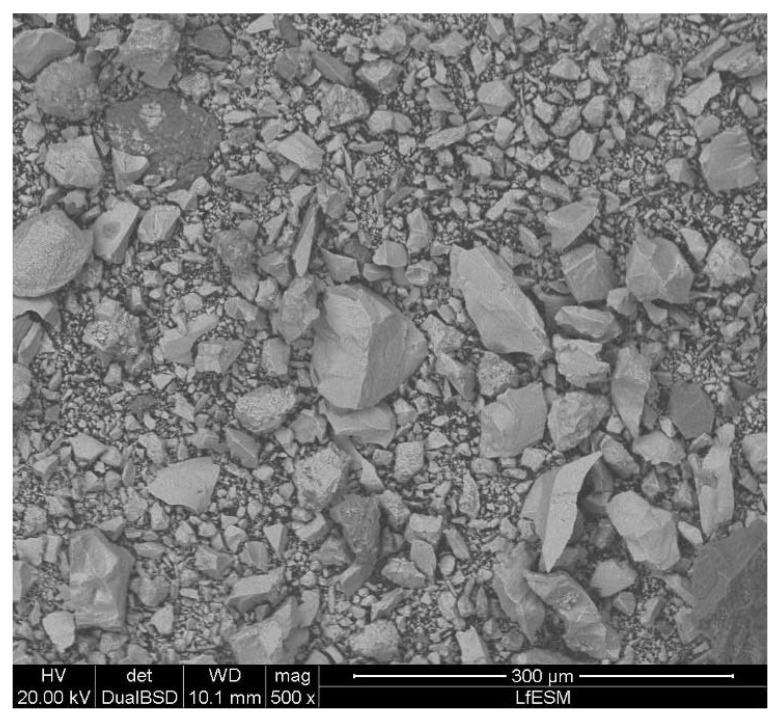
Scanning electron microscopy picture of sample E.

**Figure 3 materials-15-03943-f003:**
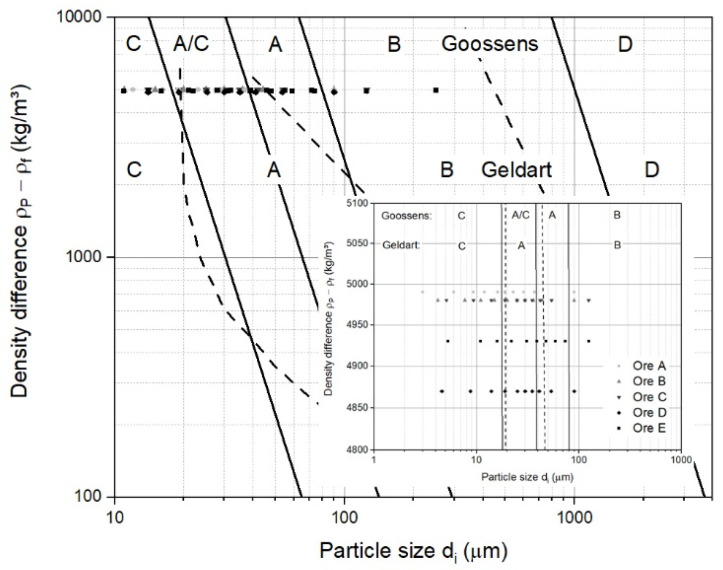
General classification diagram for fluidized particles under ambient conditions and air, including boundaries based on Geldart [40], dashed lines, and Goossens [41], solid lines, data of iron ore ultra-fines, and a detailed section.

**Figure 4 materials-15-03943-f004:**
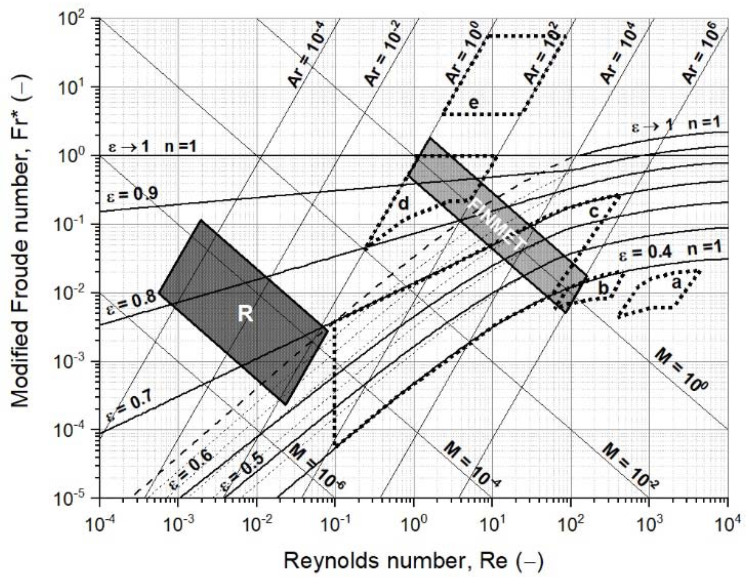
Extended version of the fluidized state diagram following Reh’s approach [48], (a) shaft furnace, (b) moving bed, (c) particulate and bubbling fluidized bed, (d) circulating fluidized bed, (e) pneumatic transport reactor, (FINMET) operating field of FINMET^®^ according to Schenk [15], and (R) operating field.

**Figure 5 materials-15-03943-f005:**
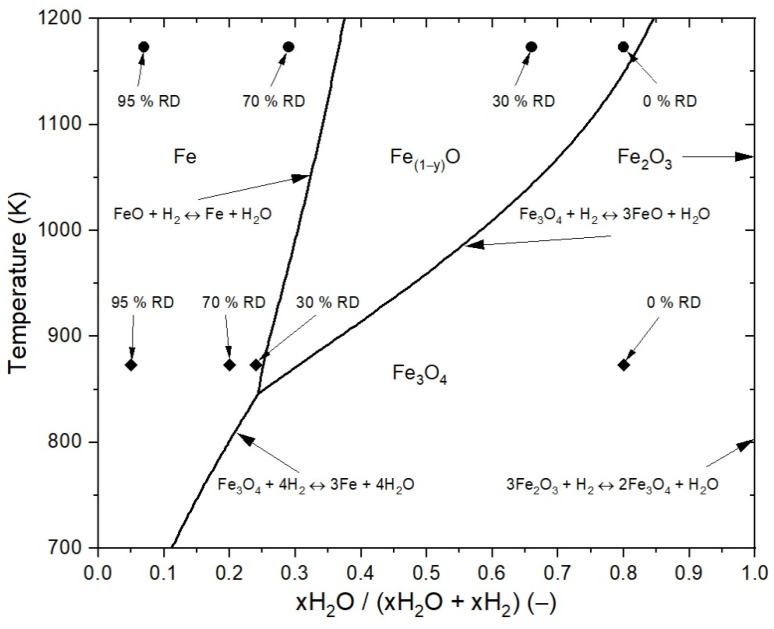
Baur–Glaessner diagram including the process conditions for 873 and 1173 K.

**Figure 6 materials-15-03943-f006:**
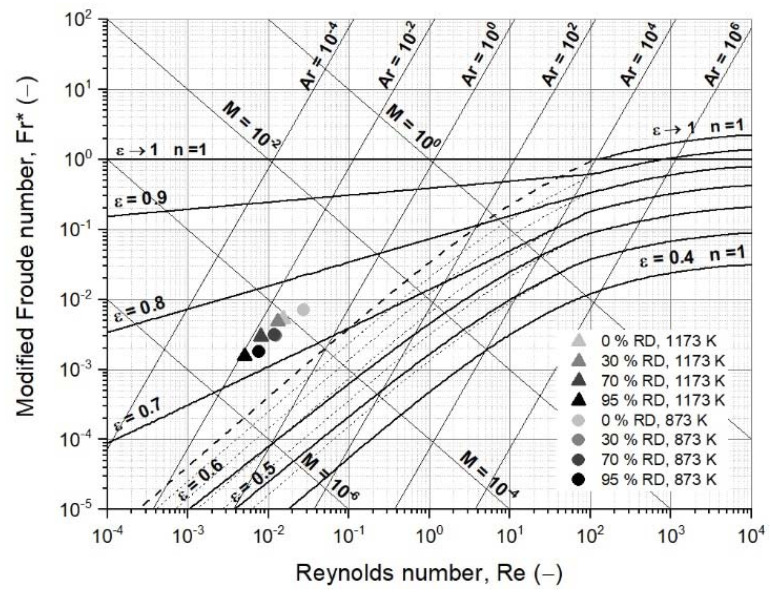
Extended fluidized state diagram following Reh’s approach, including process data [48].

**Figure 7 materials-15-03943-f007:**
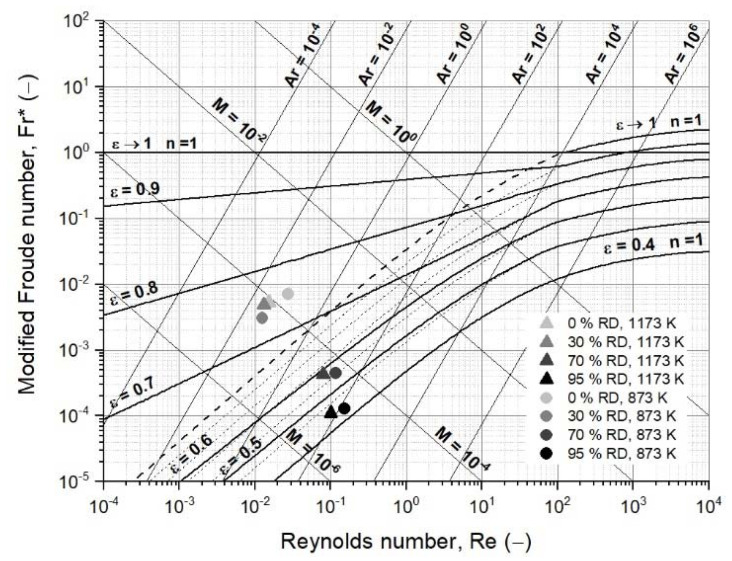
Extended fluidized state diagram following Reh’s approach, including process data for the formation of agglomerates [48].

**Figure 8 materials-15-03943-f008:**
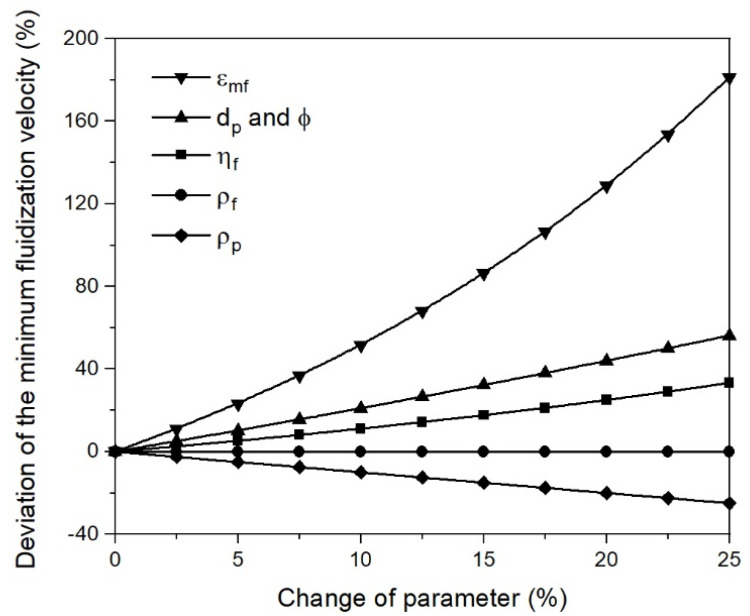
Sensitivity analysis of $u_{\mathrm{mf}}$.

**Figure 9 materials-15-03943-f009:**
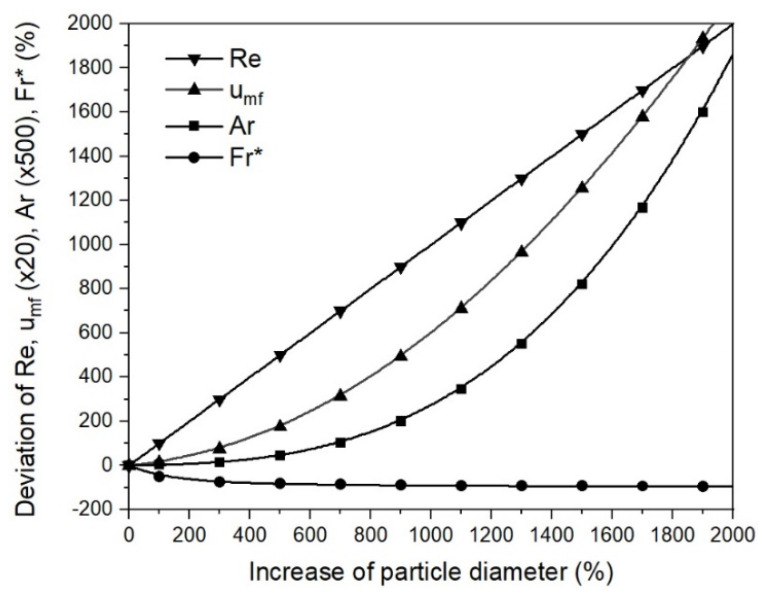
Sensitivity analysis for Re, Ar, Fr*, and $u_{\mathrm{mf}}$ from the particle diameter.

**Table 1 materials-15-03943-t001:** Material properties relating to fluidization and the chemical analysis of some typical iron ore ultra-fines.

Property	Sample A	Sample B	Sample C	Sample D	Sample E
d10 (µm)	3.0	4.2	5.1	4.6	5.3
d50 (µm)	16.1	20.0	25.0	25.2	31.1
d90 (µm)	36.9	44.0	54.2	53.5	73.8
Maximum particle size (µm)	90.0	90.0	125.0	90.0	125.0
True particle density, $\rho_{P}$ (kg/m^3^)	4990	4980	4980	4870	4930
Bulk density, $\rho_{\mathrm{Bulk}}$ (kg/m^3^)	1925	1988	2117	1996	2093
Void fraction of the bulk material, ε (−)	0.61	0.60	0.57	0.59	0.58
Sphericity, ϕ (−)	0.75	0.65	0.60	0.70	0.60
Fe_tot_ (wt.−%)	68.06	69.76	69.09	67.31	66.96
FeO (wt.−%)	23.25	30.03	27.30	27.60	25.70
SiO_2_ (wt.−%)	2.84	2.32	3.78	6.65	2.33
CaO (wt.−%)	0.13	0.15	0.01	0.01	0.93
Al_2_O_3_ (wt.−%)	1.68	0.04	0.08	0.01	0.69
MgO (wt.−%)	0.27	0.19	0.49	0.42	0.56

**Table 2 materials-15-03943-t002:** Process conditions of the mathematical case studies.

Property	0% RD	30% RD	70% RD	95% RD
Temperature, T (K)	1173/873	1173/873	1173/873	1173/873
Pressure gauge, p (barg)	0.1	0.1	0.1	0.1
Superficial gas velocity, u (m/s)	0.25	0.25	0.25	0.25
Void fraction of the bulk material, ε (−)	0.78	0.78	0.78	0.78
Slip velocity, $u_{\mathrm{slip}}$ (m/s)	0.321	0.321	0.321	0.321
Gas utilization, ζ (%)	80	80	80	20
Gas composition for 1173 K, xH_2_O (−)	0.8 (1 × 0.8)	0.66 (0.83 × 0.8)	0.29 (0.37 × 0.8)	0.07 (0.37 × 0.2)
Gas composition for 873 K, xH_2_O (−)	0.8 (1 × 0.8)	0.24 (0.30 × 0.8)	0.20 (0.25 × 0.8)	0.05 (0.25 × 0.2)

## Data Availability

The data presented in this study are available within the article.

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
