# Peer review of "Using Iron Ore Ultra-Fines for Hydrogen-Based Fluidized Bed Direct Reduction—A Mathematical Evaluation"

_materials, 2022, doi:10.3390/ma15113943_

Round 1
Reviewer 1 Report
Before the conduction of the dynamic analysis of this process, the thermodynamic feasiability should be conducted regarding to the reduction reactions by hydrogen.
1 What other fields can the fluid bed technique can be used except for the hydrogen reduction based reactions. 2 Please provide the grade of the iron ores.
Reviewer 2 Report
Please see the attachment

Reviewer 3 Report
It is a good paper nicely presented, proposing a mathematical evaluation of hydrogen-based fluidized bed using ultra-fines iron ore.
Some minor revision need to be done. When revise your paper please have in mind the following:
1. In the figure 1, the caption should explain (a) and (b) figure. If there only magnify is different one picture may be removed.
2. If the figure 5 is not the original work of the authors please insert citation.
3. In the figure 8 the caption -Sensitivity analysis of umf – should be enough. The comment-according to Ergun- may be inserted in Discussion text.
4. In the Reference chapter I strongly recommend to avoid over selfcitation. For example in this paper one of the authors appears for 5 times. Also please check again the citation format.
Round 2
Reviewer 1 Report
can be accepted.